# Framing the MHD Micropolar-Nanofluid Flow in Natural Convection Heat Transfer over a Radiative Truncated Cone

**Waqar A. Khan [1],\* , A.M. Rashad [2] , S.M.M. EL-Kabeir [3] and A.M.A. EL-Hakiem [2]**

[1]   Department of Mechanical Engineering, College of Engineering, Prince Mohammad Bin Fahd University, Al Khobar 31952, Saudi Arabia

[2]   Department of Mathematics, Aswan University, Faculty of Science, Aswan 81528, Egypt

[3]   Department of Mathematics, Prince Sattam bin Abdulaziz University, College of Science and Humanities, Al-Kharj 11942, Saudi Arabia

\*   Correspondence: wkhan@pmu.edu.sa

**Abstract:**   Recently, nanoparticles have supplied diverse challenges in the area of science. The nanoparticles suspended in several conventional fluids can convert the fluids flow and heat transmission features. In this investigation, the mathematical approach is utilized to explore the magnetohydrodynamics micropolar-nanofluid flow through a truncated porous cone. In this mathematical model, non-linear radiation and suction/injection phenomena are also scrutinized with the Tiwari-Das nanoliquid pattern. The designed system of the mathematical model of the boundary value problem is converted to a set of dimensionless non-similar equations applying convenient transformations. In this study, kerosene oil is selected as the base fluid, while the nanoparticles of $Fe_3O_4$ are utilized to promote the heat transmission rate. The problem is solved numerically using the Runge-Kutta-Fehlberg method (RKF45). It is demonstrated that an enhancement in the pertinent parameters improves the heat transmission rate.

**Keywords:** natural convection; magnetic field; thermal radiation; micropolar-nanofluid

---

## 1. Introduction

Transport of heat in several industrial and natural and operations is a fundamental and severe mechanism. Various mechanisms include several forms for the transfer of heat. However, transport of heat in the processes happening in fluid flow regimes occurs in three ways; radiation, convection, and conduction. Various strategies have been presented by scientists for the augmentation of heat transfer [1–4]. These involve surfaces being exposed to cooling, and sources, as in the case of heat exchangers, using fluids which are good conductors of heat and a blend of fluids such that the transport of heat in the blend of nanoparticles and base fluid can be promoted. Besides these classical methods, researchers working in the design of thermal systems have presented the style for the augmentation of heat by the dispersal of nanoparticles in fluids. This dispersal of nanoparticles inspires an increment in the ability to transfer of heat in the medley fluid (a medley of nanoparticles and essential fluid). Due to which, a thermal framework employing a mixture of nanoparticles and traditional fluid works as an effective technique. Due to the significance of the nanofluids in thermal frames, several studies, including numerical, experimental, and theoretical, have been performed. A pioneering study on nanofluid was undertaken by Choi [5]. It was demonstrated that the addition of a small volume fraction of nanoparticles to base fluids can increase the thermal conductivity of the fluid by up to approximately two times. Kuznetsov and Neild [6] construed natural convective flowing through an orthogonal surface saturated in nanofluid porous media. They found that the reduced Nusselt

number decreases with the controlling parameters. Gorla and Chamkha [7] found that an increase in the Nr and Nt reduces the friction factor and enhances the heat and mass transfer rates. Sameh and Mahdy [8] reviewed the natural convective flowing through a cone filled in nanofluid in the presence of a magnetic domain. They concluded that Ag-nanoparticles provide a higher heat transfer rate than $TiO_2$ nanoparticles. Oztop and Nada [9] investigated the natural convective in rectangular hot containers of a nanofluid flow and found that the volume fraction of nanoparticles, as well as the height of the heater, enhance the heat transfer rate. Reddy and Chamkha [10] studied the natural convective flowing of a magnetic field for a nanofluid past an orthogonal cone and noticed significant enhancement in the natural convection heat transfer when the size of nanoparticles decreases. Noghrehabadi et al. [11] noticed an increase in the dimensionless wall temperature with an increase in the Brownian motion parameter. Rashad et al. [12] investigated the boundary layer natural convective flowing through a vertical surface under the impact of the difference in sinusoidal temperature for a nanofluid. On the other side, the fluids that have precise construction are essential because of their entry into many industries and engineering implementations. The micropolar fluids were defined by Erignen [13,14]. Micropolar fluids belong to a category of fluids that have a non-symmetric strain sensor. These fluids consist of solid particles with random orientation that are suspended in viscous media since the distortion of the particles of fluids can be ignored. There are many implementations of micropolar fluids. Chamkha et al. [15] explained the mass and heat transport of a natural convective for micropolar fluids flow about the radiated cone. Chamkha et al. [16] investigated the magneto-natural convection flow of a micropolar fluid across an orthogonal radiated plate. Their results indicate that an increase in the micropolar fluid vortex viscosity parameter reduces the fluid linear velocity, local wall couple stress and the local Nusselt number. Bourantas and Loukopoulos [17] construed a numerical investigation of the natural convective flow past sloping canister with a square shape for a micropolar-nanofluid. They discovered that the intensity and positioning of the magnetic field substantially affect the flow and temperature fields. Khan et al. [18,19] demonstrated that all the quantities of physical interest increase along the surface of a truncated cone in Newtonian-nanofluids. Raju et al. [20] noticed that an increase in the volume fraction of ferroparticles grows the friction factor and heat transport rate. Alamgir [21] applied an integral technique to determine the overall heat transfer from vertical cones and produced a general expression for the average Nusselt number in terms of governing parameters.

The target of the current study is to discuss the influence of the magnetic field on the dispersion of nanoparticles in the micropolar fluid by natural convective flow past a porous truncated cone. Current research is a novel extension in the open literature. The simulations for Magnetohydrodynamics (MHD) micropolar-nanofluid modeled boundary value problems are carried out by employing the Runge-Kutta-Fehlberg method (RKF45) using MAPLE-19 and explorations are elucidated graphically.

## 2. Governing Equations

Suppose the steady laminar motion of MHD micropolar-nanofluid with natural convective through a truncated porous radiate cone with half-angle A. The cone surface is elastic and exposed to non-linear radiation and magnetic field phenomena. The origin of the coordinate system is located at the apex of the full cone, where $x$ appears the distance over the cone, and $y$ appears the distance perpendicular to the cone surface, as displayed in Figure 1. A continuous magnetic strength $B_0$ is utilized in orthogonal to the flow trend. The temperature of the cone is constant $T_w$ and away from the cone is considered by $T_\infty$ such that $T_w > T_\infty$. Based on these conventions, the governing boundary layer equations are stated as,

$$\frac{\partial(ru)}{\partial x} + \frac{\partial(rv)}{\partial y},\tag{1}$$

$$u\frac{\partial u}{\partial x} + v\frac{\partial u}{\partial y} = \frac{\mu_{ff} + \kappa}{\rho_{ff}}\frac{\partial^2 u}{\partial y^2} + g^*\beta_{ff}(T - T_\infty)\cos A + \frac{\kappa}{\rho_{ff}}\frac{\partial N}{\partial y} - \frac{\sigma_{ff}B_0^2}{\rho_{ff}}u,\tag{2}$$

$$u\frac{\partial N}{\partial x} + v\frac{\partial N}{\partial y} = \frac{\gamma_{ff}}{\rho_{ff}j}\frac{\partial^2 N}{\partial y^2} - \frac{\kappa}{\rho_{ff}j}\left(2N + \frac{\partial u}{\partial y}\right),$$ (3)

$$u\frac{\partial T}{\partial x} + v\frac{\partial T}{\partial y} = \frac{k_{ff}}{(\rho C_p)_{ff}}\frac{\partial^2 T}{\partial y^2} + \frac{16\sigma_1}{3(a_r + \sigma_s)(\rho C_p)_{ff}}\frac{\partial}{\partial y}\left(T^3\frac{\partial T}{\partial y}\right).$$ (4)

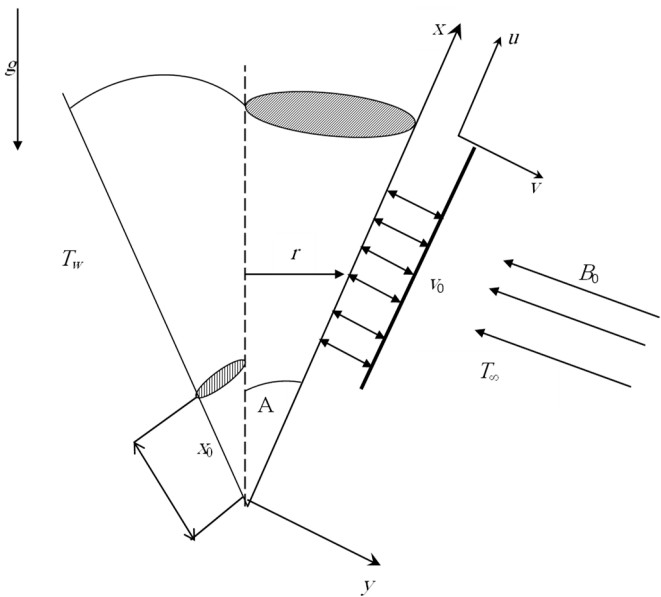

**Figure 1.** Problem Schematics and Coordinate System.

The following imposed conditions of the model are (see [22]);

$$u = 0, v = -v_0, N = 0, T = T_w : y = 0,$$ (5)

$$u \to 0, T \to 0, N \to 0, T \to T_\infty : y \to \infty,$$ (6)

where $v_0 > 0$ symbolizes the wall suction velocity; $v_0 < 0$ is the wall suction velocity injection; $v_0 = 0$ signifies that the cone surface is impermeable.

The following of dimensionless variables are used to obtain the dimensionless governing equations:

$$\xi = \frac{x - x_0}{x_0} = \frac{\overline{x}}{x_0}, \eta = \frac{y}{\overline{x}}(Gr_{\overline{x}})^{1/4}, Gr_{\overline{x}} = \frac{g\beta_f \cos A(T_w - T_\infty)\overline{x}^3}{v_f^2},$$

$$f(\xi, \eta) = \frac{\psi}{rv_f(Gr_{\overline{x}})^{1/4}}, \theta(\xi, \eta) = \frac{T - T_\infty}{T_w - T_\infty}, N = \frac{v_f(Gr_{\overline{x}})^{3/4}}{\overline{x}^2}g(\xi, \eta),$$ (7)

$$u = \frac{v_f(Gr_{\overline{x}})^{1/2}}{\overline{x}}f', v = -\frac{v_f(Gr_{\overline{x}})^{1/4}}{\overline{x}}\left[\left(\frac{\xi}{\xi + 1} + \frac{3}{4}\right)f + \xi\frac{\partial f}{\partial \xi} - \frac{1}{4}\eta f'\right]$$

Implementation of Rosseland approximation gives

$$q_r = -\frac{4\sigma_1}{3(a_r + \sigma_s)}\frac{\partial T^4}{\partial \overline{y}}.$$ (8)

In the present work, the following thermophysical relations are utilized [22];

$$\rho_{ff} = (1-\phi)\rho_f + \phi\rho_s, \mu_{ff} = \frac{\mu_f}{(1-\phi)^{2.5}}, \alpha_{ff} = \frac{k_{ff}}{(\rho C_p)_{ff}}$$
$$(\rho C_p)_{ff} = (1-\phi)(\rho C_p)_f + \phi(\rho C_p)_s, (\rho\beta)_{ff} = (1-\phi)(\rho\beta)_f + \phi(\rho\beta)_s$$
$$\frac{k_{ff}}{k_f} = (k_s + 2k_f) - 2\phi(k_f - k_s) \times \left((k_s + 2k_f) + \phi(k_f - k_s)\right)^{-1}$$
$$\frac{\sigma_{ff}}{\sigma_f} = 1 + 3\left(\frac{\sigma_s}{\sigma_f} - 1\right)\phi \times \left(\left(\frac{\sigma_s}{\sigma_f} + 2\right) - \left(\frac{\sigma_s}{\sigma_f} - 1\right)\phi\right)^{-1}$$

(9)

Here $\phi$ is nanoparticles volume fraction. The effective thermophysical properties of kerosene-based ferrofluid have been registered in Table 1. Using Equations (7)–(9), the governing equations in dimensionless form can be written as

$$\frac{\rho_f}{\rho_{ff}}\left(\frac{\mu_{ff}}{\mu_f} + K\right)f''' + \left(\frac{\xi}{\xi+1} + \frac{3}{4}\right)ff'' - \frac{1}{2}f'^2 + C_1\theta + \frac{\rho_f}{\rho_{ff}}Kg' - C_2\xi^{1/2}f' = \xi\left(f'\frac{\partial f'}{\partial\xi} - f''\frac{\partial f}{\partial\xi}\right) \quad (10)$$

$$\left(\frac{\xi}{\xi+1} + \frac{3}{4}\right)fg' - \frac{1}{4}gf' + \frac{\rho_f}{\rho_{ff}}\left(\frac{\mu_{ff}}{\mu_f} + \frac{K}{2}\right)g'' - \frac{\rho_f}{\rho_{ff}}K.B\xi^{1/2}(2g + f'') = \xi\left(f'\frac{\partial g}{\partial\xi} - g'\frac{\partial f}{\partial\xi}\right), \quad (11)$$

$$\frac{1}{\text{Pr}}\frac{\alpha_{ff}}{\alpha_f}\theta'' + \left(\frac{\xi}{\xi+1} + \frac{3}{4}\right)f\theta' + C_3\left\{\theta'[(\theta_w - 1)\theta + 1]^3\right\}' = \xi\left(f'\frac{\partial\theta}{\partial\xi} - \theta'\frac{\partial f}{\partial\xi}\right), \quad (12)$$

$$\eta = 0 : f' = 0, g = 0, f_0\xi^{1/4} = \left(\frac{\xi}{\xi+1} + \frac{3}{4}\right)f + \xi\frac{\partial f}{\partial\xi}, \theta = 1$$
$$\eta \to \infty : f' \to 0, g \to 0, \theta \to 0$$

(13)

where the constants and the dimensionless variables are defined as

$$C_1 = \frac{\rho_f}{\rho_{ff}}\frac{(\rho\beta)_{ff}}{(\rho\beta)_f}; C_2 = \frac{\rho_f}{\rho_{ff}}\frac{\sigma_{ff}}{\sigma_f}Ha^2; C_3 = \frac{4R_d}{3\text{Pr}}\frac{(\rho C_p)_f}{(\rho C_p)_{ff}}$$
$$Ha^2 = \left(\frac{\sigma_f B_0^2}{\rho_f v_f\left(x_0^4/Gr_{x_0}\right)^{1/2}}\right); f_0 = \frac{v_0}{v_f}\left(\frac{x_0^4}{Gr_{x_0}}\right)^{1/4}; B = \frac{x_0^2}{jGr_{x_o}^{1/2}}$$
$$R_d = \frac{4\sigma^* T_\infty^3}{[k_f(a_r + \sigma_s)]}; \text{Pr} = \frac{v_f}{\alpha_f}$$

(14)

**Table 1.** Thermophysical properties of base fluids and nanoparticles.

| Physical Properties | Kerosene | Fe$_3$O$_4$ |
|---|---|---|
| $C_p$ [J/kgK] | 2090 | 670 |
| $\rho$ [kg/m$^3$] | 780 | 5180 |
| $k$ [W/mK] | 0.149 | 80 |
| $\mu$ [Ns/m$^2$] | 0.00164 | - |
| $\beta$ [1/K] | $99 \times 10^{-5}$ | $20.6 \times 10^{-5}$ |
| $\sigma$ [W/mK] | $6 \times 10^{-10}$ | $0.112 \times 10^6$ |
| Pr | 23.0114 | - |

The spin-gradient nanofluid viscid $\gamma_{ff}$ is defined as

$$\gamma_{ff} = \left(\mu_{ff} + \frac{\kappa}{2}\right)j = \mu_f\left(\frac{\mu_{ff}}{\mu_f} + \frac{K}{2}\right)j, \tag{15}$$

where, $\mu_f$; the dynamic viscosity of the base fluid, and $K = \kappa/\mu_f$ is the micropolar (material) parameter. In the limiting case, it allows the governing equations to envisage the appropriate conduct when the microstructure impacts are insignificant, and the total spin lowers to the angular velocity.

Finally, the expression of local skin friction $C_f$ and local Nusselt number ($Nu_x$) are written as:

$$C_f = -2(Gr_{\overline{x}})^{-1/4}\left(\frac{\mu_{ff}}{\mu_f} + K\right)f''(\xi, 0), \ Nu_{\overline{x}} = -Gr_{\overline{x}}^{1/4}\left(\frac{k_{ff}}{k_f} + \frac{4R_d\theta_w^3}{3}\right)\theta'(\xi, 0). \tag{16}$$

It is important to note that Equations (10)–(12) are still partial differential equations as they contain the derivatives with respect to $\xi$. This is the foremost hurdle to the solution of these equations.

## 3. Local Similarity

Following References [23,24], Equations (10)–(12) are solved from the perspective of the local similarity model. In this approach, the first derivatives with respect to $\xi$ are neglected in Equations (10)–(12) and in boundary conditions (13) to obtain the following local similarity model:

$$\frac{\rho_f}{\rho_{ff}}\left(\frac{\mu_{ff}}{\mu_f} + K\right)f''' + \left(\frac{\xi}{\xi+1} + \frac{3}{4}\right)ff'' - \frac{1}{2}f'^2 + C_1\theta + \frac{\rho_f}{\rho_{ff}}Kg' + C_2\xi^{1/2}f' = 0 \tag{17}$$

$$\frac{\rho_f}{\rho_{ff}}\left(\frac{\mu_{ff}}{\mu_f} + \frac{K}{2}\right)g'' + \left(\frac{\xi}{\xi+1} + \frac{3}{4}\right)fg' - \frac{1}{4}gf' + \frac{\rho_f}{\rho_{ff}}K.B\xi^{1/2}(2g + f'') = 0 \tag{18}$$

$$\frac{1}{Pr}\frac{\alpha_{ff}}{\alpha_f}\theta'' + \left(\frac{\xi}{\xi+1} + \frac{3}{4}\right)f\theta' + \frac{4R_d}{3Pr}\frac{(\rho C_p)_f}{(\rho C_p)_{ff}}\left\{\theta'[(\theta_w - 1)\theta + 1]^3\right\}' = 0 \tag{19}$$

$$\begin{aligned} \eta = 0: f' = 0, g = 0, f_0\xi^{1/4} = \left(\frac{\xi}{\xi+1} + \frac{3}{4}\right)f, \theta = 1 \\ \eta \to \infty: f' \to 0, g \to 0, \theta \to 0 \end{aligned} \tag{20}$$

Equations (17)–(19) are Ordinary differential equations (ODEs) and can be solved numerically with boundary conditions (20) applying the Runge-Kutta-Fehlberg method (RKF7 45).

## 4. Local Non-Similarity Solution Method

Following [23,24], we retain all the terms by assuming the new auxiliary functions $F(\xi, \eta)$, $G(\xi, \eta)$, $\Theta(\xi, \eta)$ which are defined by

$$F = \frac{\partial f}{\partial \xi}, G = \frac{\partial g}{\partial \xi}, \Theta = \frac{\partial \theta}{\partial \xi}. \tag{21}$$

Using these functions, Equations (10)–(12) can be re-written as

$$\frac{\rho_f}{\rho_{ff}}\left(\frac{\mu_{ff}}{\mu_f} + K\right)f''' + \left(\frac{\xi}{\xi+1} + \frac{3}{4}\right)ff'' - \frac{1}{2}f'^2 + C_1\theta + \frac{\rho_f}{\rho_{ff}}Kg' - C_2\xi^{1/2}f' = \xi(f'F' - f''F) \tag{22}$$

$$\left(\frac{\xi}{\xi+1} + \frac{3}{4}\right)fg' - \frac{1}{4}gf' + \frac{\rho_f}{\rho_{ff}}\left(\frac{\mu_{ff}}{\mu_f} + \frac{K}{2}\right)g'' - \frac{\rho_f}{\rho_{ff}}K.B\xi^{1/2}(2g + f'') = \xi(f'G - g'F) \tag{23}$$

$$\frac{1}{Pr}\frac{\alpha_{ff}}{\alpha_f}\theta'' + \left(\frac{\xi}{\xi+1} + \frac{3}{4}\right)f\theta' + C_3\left\{\theta'[(\theta_w - 1)\theta + 1]^3\right\}' = \xi(f'\Theta - \theta'F), \tag{24}$$

with the boundary conditions

$$
\eta = 0 : f' = 0, g = 0, f_0\xi^{1/4} = \left(\tfrac{\xi}{\xi+1} + \tfrac{3}{4}\right)f + \xi F, \theta = 1
$$
$$
\eta \to \infty : f' \to 0, g \to 0, \theta \to 0 \tag{25}
$$

Equations (22)–(24) with boundary conditions (25) exemplify a local nonsimilarity model for the current problem. To obtain a local similarity model, Equations (21)–(24) are differentiated w.r.t. $\xi$, simplified, and the first-order derivatives w.r.t. $\xi$ are neglected again. These equations are given by

$$
\frac{\rho_f}{\rho_{ff}}\left(\frac{\mu_{ff}}{\mu_f} + K\right)F''' \quad +\left(\tfrac{\xi}{\xi+1} + \tfrac{3}{4}\right)(Ff'' + fF'') - 2f'F' + C_1\Theta + \frac{\rho_f}{\rho_{ff}}KG'
$$
$$
-C_2\left(\frac{1}{2\xi^{1/2}}f' - \xi^{1/2}F'\right) + Ff'' - \xi\left(F'^2 - F''F\right) = 0 \tag{26}
$$

$$
\frac{\rho_f}{\rho_{ff}}\left(\frac{\mu_{ff}}{\mu_f} + \frac{K}{2}\right)G'' - \frac{\rho_f}{\rho_{ff}}K.B\left[\frac{(2g + f'')}{2\xi^{1/2}} + \xi^{1/2}(2G + F'')\right] + \left(\frac{1}{\xi+1} - \frac{\xi}{(\xi+1)^2}\right)fg' +
$$
$$
\left(\frac{\xi}{\xi+1} + \frac{3}{4}\right)(Fg' + fG') - \frac{1}{4}(5Gf' + gF') + g'F - \xi(F'G - G'F) = 0 \tag{27}
$$

$$
\tfrac{1}{Pr}\tfrac{\alpha_{ff}}{\alpha_f}\Theta'' + \left(\tfrac{1}{\xi+1} - \tfrac{\xi}{(\xi+1)^2}\right)f\theta' + \left(\tfrac{\xi}{\xi+1} + \tfrac{3}{4}\right)(F\theta' + f\Theta') + Rd\cdot C_3\Theta''\left[(\theta_w - 1)\theta + 1\right]^3 + 3\theta''\left[(\theta_w - 1)\theta + 1\right]^2
$$
$$
(\theta_w - 1)\Theta + 6\theta'\left[(\theta_w - 1)\theta + 1\right]^2(\theta_w - 1)\Theta' + 6\theta'^2\left[(\theta_w - 1)\theta + 1\right](\theta_w - 1)\Theta - \xi(F'\Theta - \theta'F) = 0 \tag{28}
$$

with the boundary conditions

$$
\eta = 0 : F' = 0, G = 0, f_0\xi^{1/4} = \left(\tfrac{\xi}{\xi+1} + \tfrac{3}{4}\right)F, \theta = 1
$$
$$
\eta \to \infty : F' \to 0, G \to 0, \Theta \to 0 \tag{29}
$$

Equations (22)–(24) and (26)–(28) with boundary conditions (25) and (29) were solved numerically by employing the Runge-Kutta-Fehlberg method (RKF45) using MAPLE-19 software. RKF 45 is a technique of order $O(h^4)$ with a fault estimator of order $O(h^5)$. This method is generally known as one of the "best methods" available for solving a system of nonlinear differential equations and supply the most effective data. Step size $\Delta\eta = 0.001$ and a convergence criterion of $10^{-6}$ were selected in the numerical computations. The asymptotic boundary conditions, given in Equations (25) and (29), were replaced by using a value of 12 for the similarity variable $\eta_{max}$ as follows:

$$
f'(12) = 0, g(12) = 0, \theta(12) = 0, F'(12) = 0, G(12) = 0, \Theta(12) = 0. \tag{30}
$$

The selection of $\eta_{max} = 12$ ensures that all numerical solutions approached the asymptotic values properly. The other details of this method can be found in Reference [25].

## 5. Discussions

The set of nonlinear coupled ordinary differential Equations (22)–(24) and (26)–(28) subjected to boundary conditions (25) and (29) are solved numerically using the RKF 45 method. The influence of pertinent parameters on the dimensionless velocities and temperature, along with the friction factor coefficient and local Nusselt numbers, are analyzed and explored graphically.

The effects of the micropolar parameter on dimensionless velocity are depicted in Figure 2a for suction, Figure 2b for impermeable, and Figure 2c for injection of kerosene oil-based magnetite nanofluid over a truncated cone. In this case, the micropolar parameter boosts from $K = 0$ (Newtonian fluid) to $K = 2$ (micropolar fluid). In each case, the velocity is plotted at different stations along the truncated cone. With the cone wall, the dimensionless velocity shoots up and then declines in the hydrodynamic boundary layer. It is noted that the velocity is the maximum for a Newtonian fluid ($K = 0$) and lowers for the micropolar fluid ($K = 2$) in each situation. The comparison shows that the maximum velocity is lowest in the case of suction ($f_0 > 0$) and is highest for the injection case

($f_0 < 0$), see Figure 2c. The difference of the temperature with the micropolar parameter K for the three cases is depicted in Figure 3a–c inside the thermal boundary-layer. For the suction, no appreciable effect could be observed in the temperature distribution of both Newtonian ($K = 0$) and micropolar fluids ($K = 2$). However, when the cone surface is impermeable or solid, the thermal boundary layer is found to be thinner for the Newtonian fluid than the micropolar fluid, see Figure 3b. Into the thermal boundary-layer, the dimensionless temperature of both fluids increases through the surface. When the fluids are injected into the cone surface, the dimensionless temperature remains the same across the surface and then decreases exponentially up to the ambient temperature. Again, the dimensionless temperature converges quickly for the Newtonian fluid, see Figure 3c. Figure 4a–c present the impacts of micropolar parameter on the dimensionless rotating velocity along the cone surface for the three selected cases. As expected, there is no rotating velocity of the Newtonian fluid ($K = 0$) in any case. However, for the micropolar fluid ($K = 2$), the behavior of rotating fluid can be witnessed in each case, see Figure 4a–c. This is due to the rotation of the rigid particles present in the micropolar fluids. These fluids not only support body couples but also explain micro-rotational impacts. The comparison shows the lowest value of the maximum rotational velocity for the suction case and the highest value for the injection of micropolar fluid.

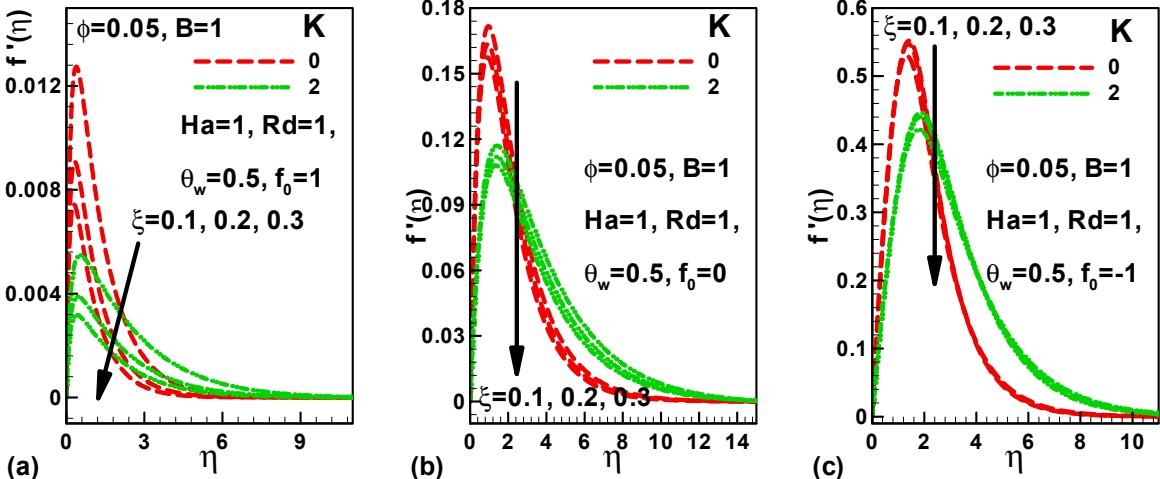

**Figure 2.** Effects of micropolar parameter on dimensionless velocity along truncated cone for (**a**) suction, (**b**) impermeable, and (**c**) injection of kerosene-based magnetite nanofluid.

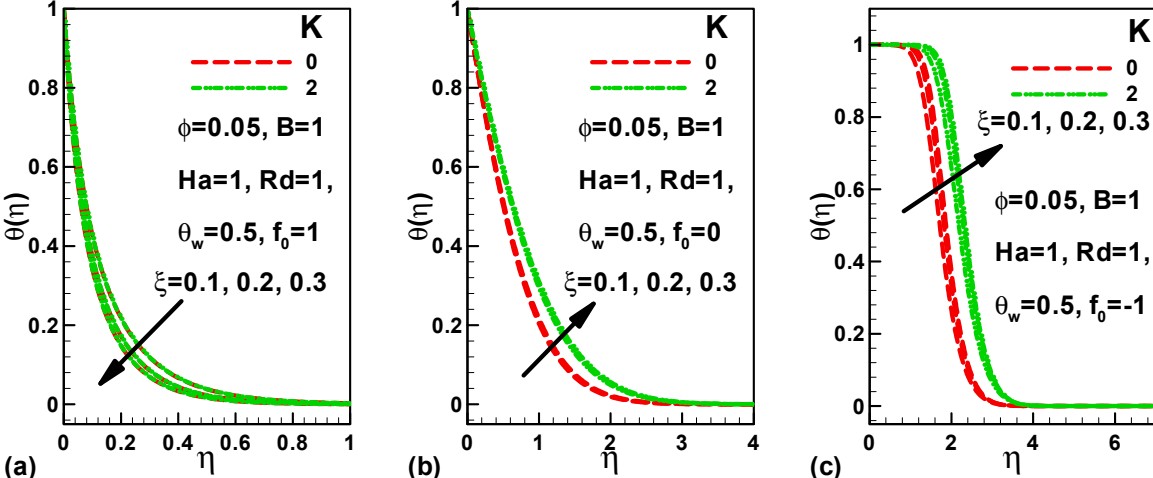

**Figure 3.** Effects of micropolar parameter on dimensionless temperature along truncated cone for (**a**) suction, (**b**) impermeable, and (**c**) injection of kerosene-based magnetite nanofluid.

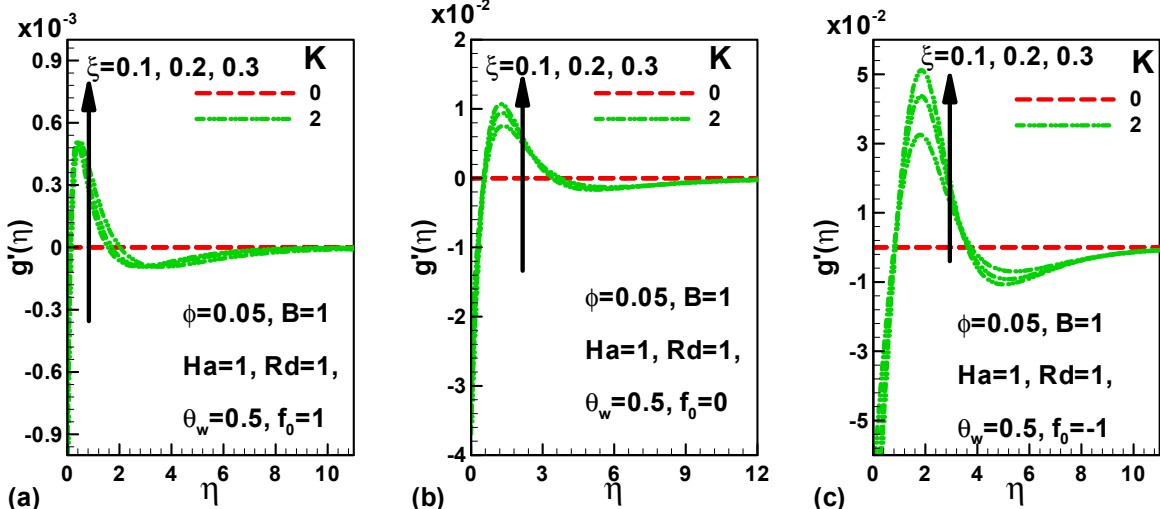

**Figure 4.** Effects of micropolar parameter on dimensionless rotating velocity along truncated cone for
(**a**) suction, (**b**) impermeable, and (**c**) injection of kerosene-based magnetite nanofluid.

Figures 5–7 exhibit the impacts of the magnetic parameter and the solid volume fraction of
nanoparticles of $Fe_3O_4$ on the dimensionless velocity, temperature, and rotating velocity, respectively.
In the suction case, no appreciable impact of the magnetic parameter could be observed for the pure
fluid ($\phi = 0$), Figure 5a. However, as the solid volume fraction of nanoparticles boosts, the impact of the
magnetic parameter can be noticed. Due to the higher density of nanoparticles and Lorentz forces, the
velocity of the fluid reduces to zero in the hydrodynamic boundary layer. The boundary-layer thickness
depends upon the magnetic parameter as well as the solid volume fraction. The same behavior of the
velocity profiles can be observed in the case of an impermeable surface, see Figure 5b. However, in
this case, the hydraulic resistance will be more significant due to the larger boundary-layer thickness.
Again, in the injection case, no visible effect of the magnetic parameter could be observed for the pure
fluid ($\phi = 0$), see Figure 5c. This effect is more pronounced when the solid volume fraction increases.
The qualitative behavior of velocity profiles is the same as in previous cases. However, the thickness of
the hydrodynamic boundary-layer is found to be the largest in the case of injection.

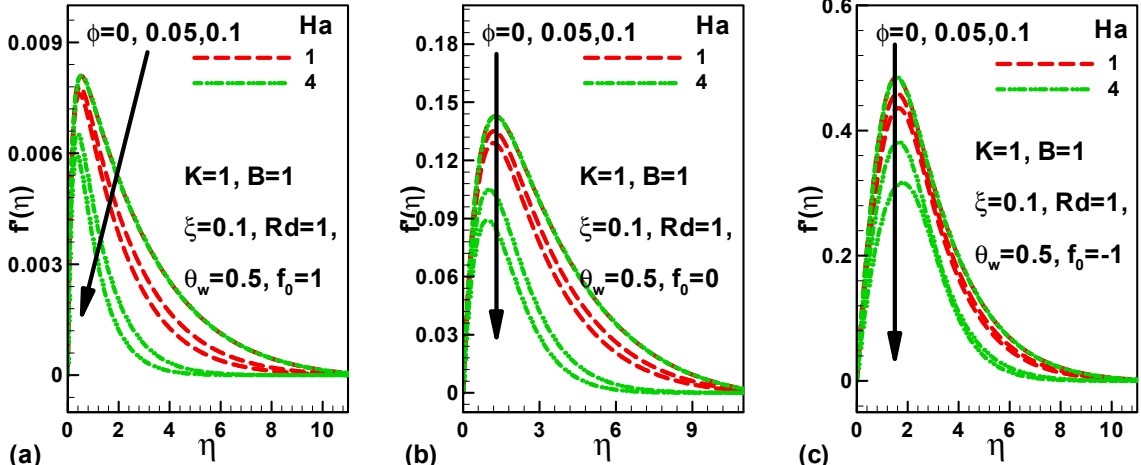

**Figure 5.** Effects of magnetic parameter and solid volume fraction of nanoparticles on dimensionless
velocity along truncated cone for (**a**) suction, (**b**) impermeable, and (**c**) injection of kerosene-based
magnetite nanofluid.

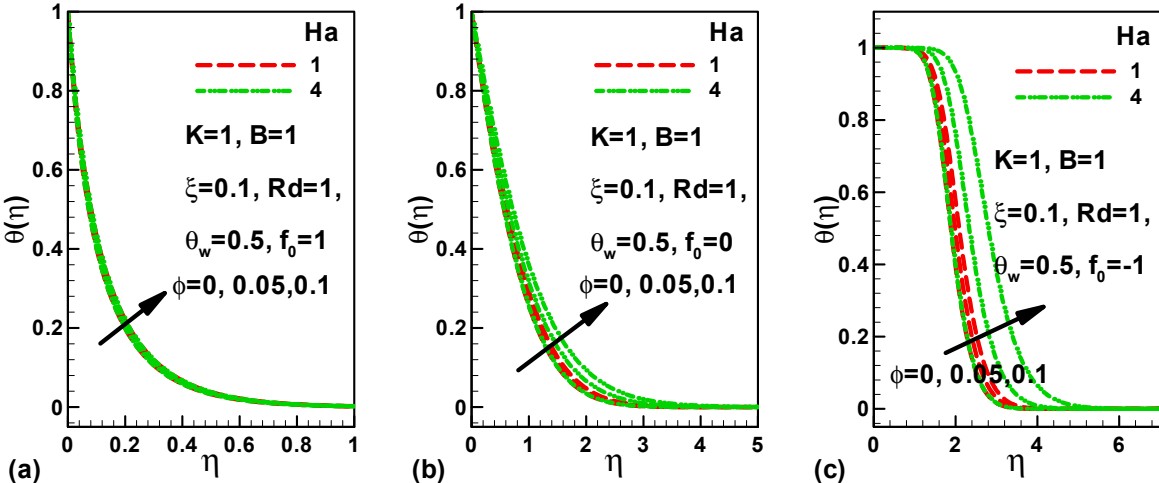

**Figure 6.** Effects of magnetic parameter and solid volume fraction of nanoparticles on dimensionless temperature along truncated cone for (**a**) suction, (**b**) impermeable, and (**c**) injection of kerosene-based magnetite nanofluid.

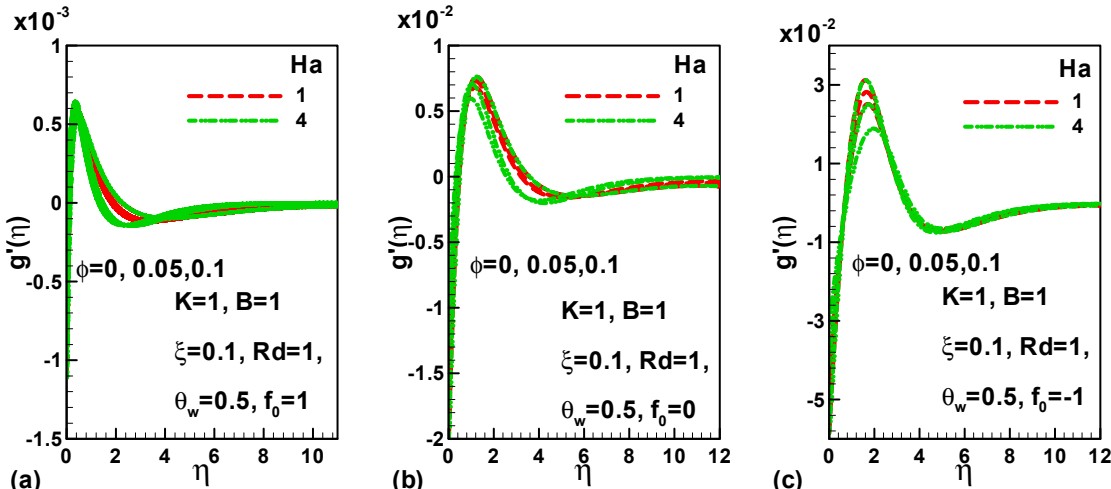

**Figure 7.** Effects of magnetic parameter and solid volume fraction of nanoparticles on dimensionless rotating velocity along truncated cone for (**a**) suction, (**b**) impermeable, and (**c**) injection of kerosene-based magnetite nanofluid.

Figure 6a demonstrates no visible impact of the magnetic parameter and the solid volume fraction on the temperature of micropolar fluid in case of suction. All the temperature profiles converge quickly inside the thermal boundary-layer, as seen in Figure 6a. Similarly, in the case of an impermeable surface, the temperature profiles converge together and show no influence of $\phi$ on the temperature. However, when the cone surface is impermeable, the impact of solid volume fraction is visible for higher values of the magnetic parameter, see Figure 6b. In this case, the thermal boundary-layer thickness is found to be larger than the suction case. In the injection case, the temperature remains constant in the neighborhood of the wall cone and then decreases exponentially in the thermal boundary layer up to the ambient temperature, as seen in Figure 6c. The thermal boundary-layer thickness boosts with the growth in the solid volume fraction. The nature of the dimensionless rotating velocity along the cone surface is shown in Figure 7a–c for the three selected cases. The rotating velocity increases sharply in the neighborhood of the cone surface and shows the oscillatory nature. The comparison shows that the maximum amplitude increases from suction to injection.

The variation in the skin friction along the cone surface is depicted in Figure 8a–c for different values of micropolar and magnetic parameters. In each case, the skin friction decreases along the cone surface with the magnetic parameter but increases with the micropolar parameter. It is well known that the magnetic field creates Lorentz forces, which tend to reduce the velocity of both fluids. In the case of micropolar fluids, this effect is more pronounced due to the translational and rotational motion of the particles.

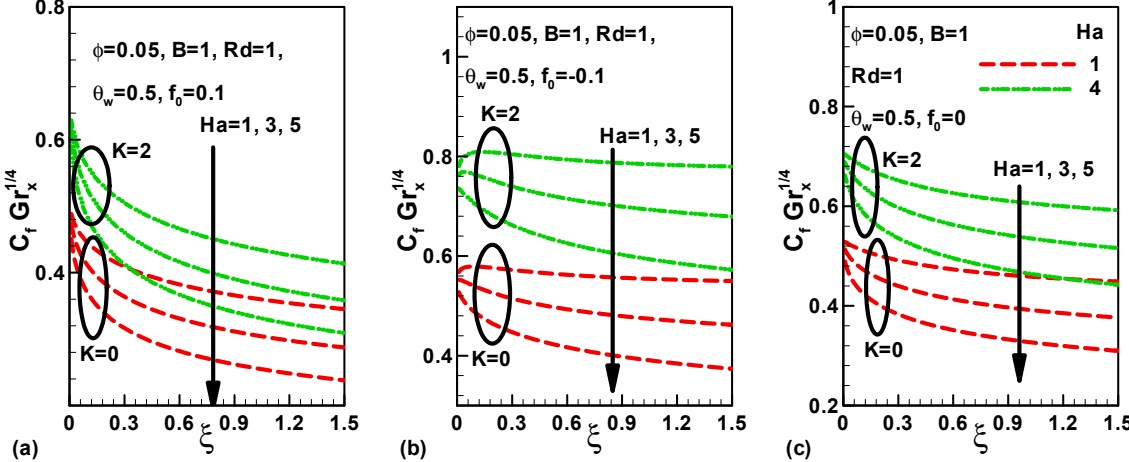

**Figure 8.** Effects of magnetic and micropolar parameters on skin friction along truncated cone for (**a**) suction, (**b**) injection, and (**c**) impermeable of kerosene-based magnetite nanofluid.

Consequently, the skin friction of micropolar fluids is found to be higher in each case. The comparison shows that the skin friction of both fluids is less in suction than in injection. Figure 9a–c display the difference in skin friction of micropolar fluids with the solid volume fraction for different values of radiation and wall temperature parameters with the magnetic impact. Due to local structure, micromotion, and higher density of the nanoparticles, the dimensionless velocity decreases and, as a result, the skin friction decreases with $\phi$ in all cases. However, radiation and wall temperature parameters help in increasing the skin friction in each case. It is demonstrated that the skin friction is always lowest in the case of suction and highest in case of injection.

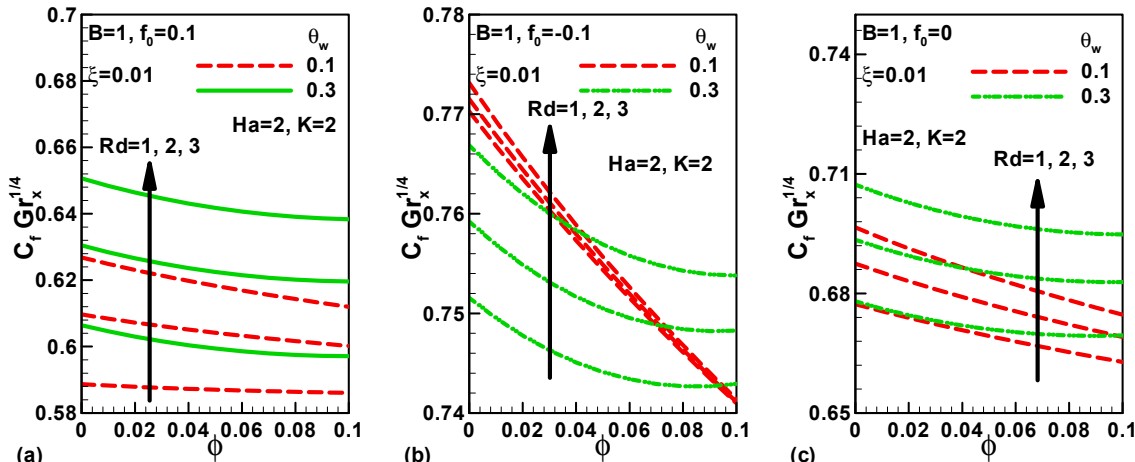

**Figure 9.** Variation of skin friction with solid volume fraction of nanoparticles, radiation, and wall temperature parameters along truncated cone for (**a**) suction, (**b**) injection, and (**c**) impermeable of kerosene-based magnetite nanofluid.

Due to local structure and micromotion of the fluid particles, micropolar fluids demonstrate specific microscopic effects on the heat transfer. These effects are compared in Figure 10a–c for Newtonian and micropolar fluids with the magnetic impact. The Nusselt numbers of Newtonian fluids are found to be higher than the Nusselt numbers of micropolar fluids in each case. In the suction and injection cases, the Nusselt numbers increase along the cone surface as shown in Figure 10a,b. However, an increase in the magnetic strength declines the Nusselt number in each case. The variation of the Nusselt number of micropolar fluid with $\phi$ is portrayed in Figure 11a for suction, in Figure 11b for injection, and in Figure 11c for the impermeable surface of the cone. This variation is shown for several values of radiation and wall temperature parameters. It is seen that the Nusselt number enhances with the increment in the solid volume fraction and the radiation parameter in each case. However, the difference in the Nusselt number with wall temperature depends upon the properties of the micropolar fluids.

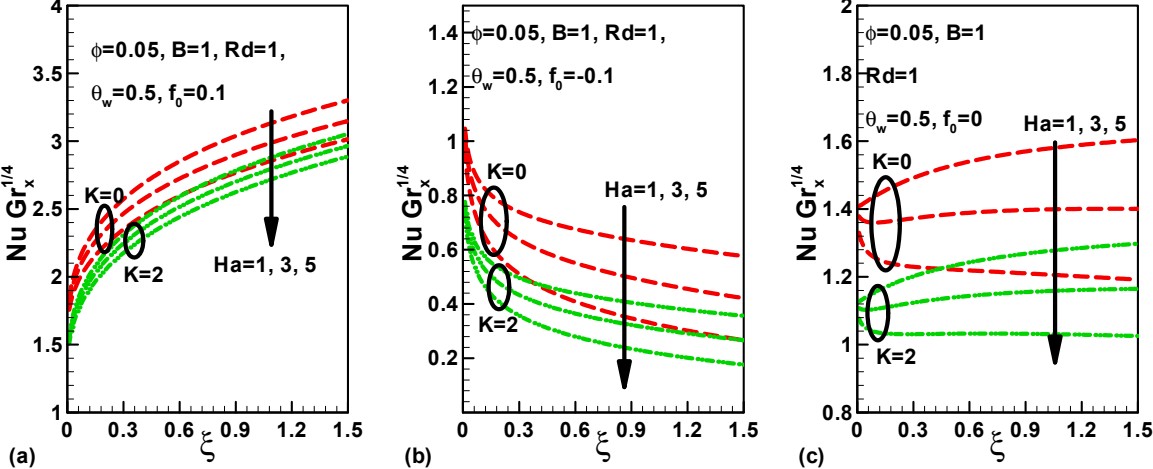

**Figure 10.** Effects of magnetic and micropolar parameters on Nusselt number along truncated cone for (**a**) suction, (**b**) injection, and (**c**) impermeable of kerosene-based magnetite nanofluid.

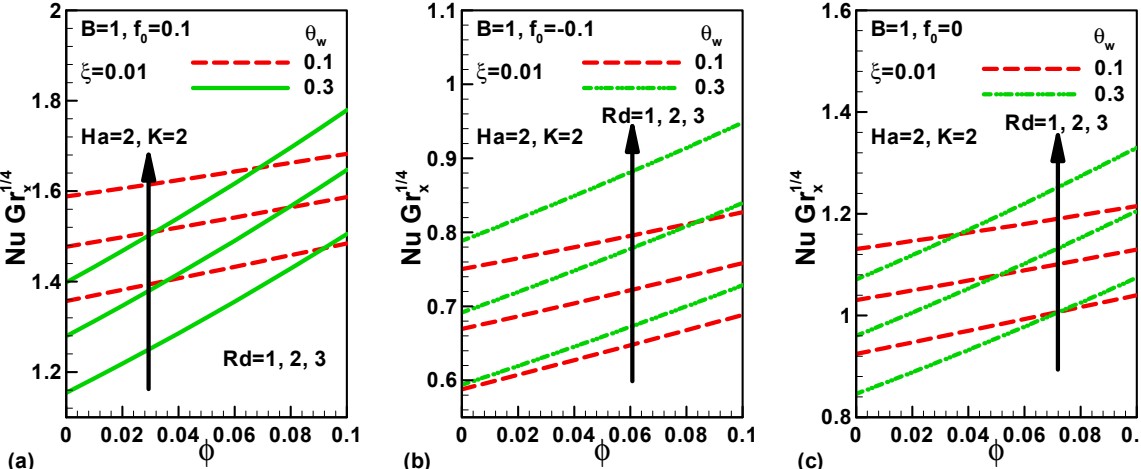

**Figure 11.** Variation of Nusselt number with solid volume fraction of nanoparticles, radiation, and wall temperature parameters along truncated cone for (**a**) suction, (**b**) injection, and (**c**) impermeable of kerosene-based magnetite nanofluid.

## 6. Conclusions

This study investigates the effects of the governing parameters on the dimensionless velocities, temperature, local skin friction, and Nusselt numbers in MHD natural convection. The dimensionless

local non-similar equations were solved numerically using the Runge-Kutta-Fehlberg method (RKF45). The following conclusions were drawn from this investigation:

- Increasing the solid volume fraction of nanoparticles, micropolar, and magnetic parameters raise the dimensionless temperature in injection and impermeable surfaces.
- Increasing solid volume fraction, micropolar and magnetic parameters reduce the dimensionless velocity in each case.
- Increasing micropolar and radiation parameters increase local skin friction, whereas increasing the magnetic field and solid volume fraction reduce the local skin friction.

Increasing the solid volume fraction of nanoparticles, radiation, micropolar, and magnetic parameters enhance the Nusselt number.

**Author Contributions:** W.A.K.: Conceptualization, Methodology, Results and discussion, Software; A.M.R.: Formulation, Funding acquisition, Writing- Original draft preparation; S.M.M.E.-K.: Visualization, Funding acquisition, Project Management; A.M.A.E.-H.: Software, Validation, Reviewing and Editing. All authors have read and agreed to the published version of the manuscript.

**Funding:** This research received no external funding.

**Conflicts of Interest:** The authors declare no conflicts of interest.

## Nomenclature

| | |
|---|---|
| A | half angle of the truncated cone |
| $a_r$ | Rosseland mean extinction coefficient |
| B | spin gradient viscosity parameter |
| $B_0$ | Magnetic inducement |
| $C_p$ | Specific heat at constant pressure [Jkg$^{-1}$·K$^{-1}$] |
| $c_f$ | Local skin-friction coefficient |
| f | dimensionless stream function |
| $f_0$ | Wall mass transfer coefficient |
| g | Dimensionless angular velocity |
| $g^*$ | Gravitational acceleration [ms$^{-2}$] |
| $Gr_{\overline{x}}$ | Local Grashof number |
| Ha | Hartmann number |
| j | microinertia density |
| k | Thermal conductivity [Wm$^{-1}$K$^{-1}$] |
| K | Dimensionless material parameter |
| N | angular velocity |
| Nu | Local Nusselt number |
| Pr | Prandtl number |
| *r* | local radius of the truncated cone [m] |
| $R_d$ | radiation-conduction parameter |
| T | Temperature of the fluid in the boundary layer [K] |
| $T_\infty$ | Temperature of the ambient fluid [K] |
| u | velocity component along x-direction [ms$^{-1}$] |
| $U_r$ | reference velocity [ms$^{-1}$] |
| v | velocity component along y-direction [ms$^{-1}$] |
| $v_0$ | wall suction or injection velocity [ms$^{-1}$] |
| x | stream wise coordinate [m] |
| $x_0$ | distance of the leading edge of truncated cone measured from the origin [m] |
| $\overline{x}$ | distance measured from the leading edge of the truncated cone, $x - x_0$ [m] |
| y | transverse coordinate [m] |

**Greek symbols**

| | |
|---|---|
| $\alpha$ | thermal diffusivity [$m^2s^{-1}$] |
| $\beta$ | thermal expansion coefficient |
| $\eta$ | pseudo similarity variable |
| $\gamma$ | spin gradient viscid |
| $\phi$ | nanoparticles volume fraction |
| $\mu$ | dynamic viscosity (Pa.s) |
| $\nu$ | kinematic viscosity ($m^2s^{-1}$) |
| $\theta$ | dimensionless temperature |
| $\theta_w$ | surface temperature parameter |
| $\rho$ | density (kg m$^{-3}$) |
| $\sigma$ | electrical conductivity (S m$^{-1}$) |
| $\sigma_1$ | Stefan-Boltzmann constant |
| $\sigma_s$ | scattering coefficient |
| $\psi$ | stream function |
| $\xi$ | dimensionless distance |

**Subscripts**

| | |
|---|---|
| f | Base fluid |
| ff | Ferrofluid |
| s | Solid nanoparticle |

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
