# Peer review of "Framing the MHD Micropolar-Nanofluid Flow in Natural Convection Heat Transfer over a Radiative Truncated Cone"

_processes, doi:10.3390/pr8040379_

Round 1

Reviewer 1 Report

Comments for the authors:

In this work, the study of the impact of the magnetic field on the dispersion of nanoparticles in the micropolar fluid by natural convective flow past a porous truncated cone is performed. By means of the Runge-Kutta-Fehlberg method, the simulations for MHD micropolar-nanofluid modeled boundary value problems have allowed to obtain the conclusions following : 1- Increasing the solid volume fraction of nanoparticles, micropolar, and magnetic parameters raise the dimensionless temperature in injection and impermeable surfaces. 2- Increasing solid volume fraction of nanoparticles, micropolar and magnetic parameters reduce the dimensionless velocity in each case. 3- Increasing micropolar and radiation parameters increase local skin friction, whereas increasing the magnetic field and solid volume fraction of nanoparticles reduce the local skin friction.

The reviewer believes that the current version of the paper is not suitable for publication in Water on the following reasons:

1- In the section 1- Introduction : the bibliographical references are good but the authors do not synthesize the main results of these bibliographical references. It would be interesting to present these existing results in the scientific literature to better show the contribution of this work.

2- In the section 2- Governing equations:

To better understand the equations (from 1 to 4) presented in this section, the definition and unit of each variable in these equations should be added.

To better understand the dimensionless variables in the equation (6), the physical meaning should be added, for example Gr. I have the same comment for the equations (13) (for example Ha) and (15) (for example Nux).

Can you recall the validation domain where the Rosseland approximation may be used ? I have the same question for the thermophysical relations given in the equation (8).

3- In the section 3- Local similarity :

Can you explain the simplifications used to obtain the equations (from 16 to 19) ?

Can you explain why you have used the Runge-Kutta-Fehlberg method for solving the system of equations (from 16 to 19) ?

4- In the section 4- Local Non-similarity Solution Method:

At the line 158, how have you fixed the step size 0.001 and the convergence criterion of 6 10^-6 ? A convergence study should be performed in order to clarify this point.

At the lines 162 and 163, the authors say that the selection of hmax 12 ensures that all numerical solutions approached the asymptotic values properly. The other details of this method can be found in [25]. Is it possi

5- In the section 5- Results and discussion:

To better understand the analysis of results, can you recall the definition of the local Nusselt number ?

Can you improve the quality of figures ?

Can you make the link between the arrows and circles on the figures and the analysis of these figures in the section 5- Results and discussion ?

6- In the section 6- Conclusions

Do the conclusions remain valid for other nanoparticles?

What are the perspectives of this work ?

Author Response

We have reduced the similarity index less than 15% and responded to the reviewer point by point.

Reviewer 2 Report

The authors use a mathematical model in order to describe the influence of solid volume fraction of nanoparticles, radiation, micropolar, and magnetic parameters on dimensionless temperature, velocity, skin friction and Nusselt number of a nanofluid flow over a radiative truncated cone.

The manuscript is well described and discussed and I would recommend to accept the manuscript for publication in proceses.

Author Response

Thanks for the nice comments and accepting the paper for publication in the journal processes.

Round 2

Reviewer 1 Report

Three major comments :

3- In the section 3- Local similarity : Can you explain the simplifications used to obtain the equations (from 16 to 19) ? Can you explain why you have used the Runge-Kutta-Fehlberg method for solving the system of equations (from 16 to 19) ? 

Response: In Eqns. (16)-(19), no simplification is required. Just you need to neglect the first order derivatives with respect to in Eqns. (9)-(11) and in boundary conditions (12). We have employed Runge-Kutta-Fehlberg method (RKF45) because this method is generally known as one of the "best
methods" available for solving a system of nonlinear differential equations and provides the most efficient and accurate results.

Is it possible to add a convergence study for the validation in your paper ?

4- In the section 4- Local Non-similarity Solution Method: At the line 158, how have you fixed the step size 0.001 and the convergence criterion of 6 10^-6 ? A convergence study should be performed in order to clarify this point. At the lines 162 and 163, the authors say that the selection of hmax 12 ensures that all numerical solutions approached the asymptotic values properly. The other details of this method can be found in [25].

Response: The step size  =  0.001 and the convergence criterion of
6 10− were selected after an extensive study of the numerical computations. The selection of max also depends upon these numerical
computations.

Is it possible to add a convergence study for the validation in your paper ?

6- In the section 6- Conclusions Do the conclusions remain valid for other nanoparticles? What are the perspectives of this work?

Response: The conclusions are valid for other nanoparticles depending upon the thermal conductivity of the particles. The results of this work can be used in several industrial and engineering applications like in microelectronics, fuel cells, hybrid-powered engines nuclear reactors,transportations, biomedicine/ pharmaceutical processes and pasteurization of food etc.

Can you change your conclusions by adding this response ?

Author Response

Reviewer 1:

Comments and Suggestions for Authors

Three major comments :

3- In the section 3- Local similarity : Can you explain the simplifications used to obtain the equations (from 16 to 19) ? Can you explain why you have used the Runge-Kutta-Fehlberg method for solving the system of equations (from 16 to 19) ? 

Response: We have adopted standard method as explained in [23, 24]. In this method, the authors have mentioned solution in two steps. In the first step, we must ignore the first order partial derivatives with respect to x to obtain similar equations. These similar equations can be solved by any suitable numerical method. Using this step, we obtained Eqns. (16)-(19). Then in the second step, we must assume new auxiliary functions and re-write the original equations. Then differentiate these equations again with respect to x and ignore again the first order partial derivatives with respect to x to obtain similar equations.   Using this step, we obtained Eqns. (25)-(28). We have employed Runge-Kutta-Fehlberg method (RKF45) because this method is generally known as one of the "best methods" available for solving a system of nonlinear differential equations and provides the most efficient and accurate results.

Is it possible to add a convergence study for the validation in your paper?

4- In the section 4- Local Non-similarity Solution Method: At the line 158, how have you fixed the step size 0.001 and the convergence criterion of 6 10^-6 ? A convergence study should be performed in order to clarify this point. At the lines 162 and 163, the authors say that the selection of hmax 12 ensures that all numerical solutions approached the asymptotic values properly. The other details of this method can be found in [25].

Response: In the software MAPLE-19, the convergence study is included. It checks by itself and informs about the convergence. If the solution is not converged, we will get no result and we must change the step size, no. of iterations or mesh size. After many trials, we selected the step size Dh=0.001, hmax=12 and the convergence criterion of 10−6. The selection of hmax also depends upon these numerical computations.

Is it possible to add a convergence study for the validation in your paper?

Response: MAPLE is different with ANSYS or FLUENT, where we get the results for different number of grids and the results converge after certain number of grids. In case of MAPLE, we can not get any result, if the solution is not converged. So, it is not possible to add a convergence study in the paper.

6- In the section 6- Conclusions Do the conclusions remain valid for other nanoparticles? What are the perspectives of this work?

Response: For other nanoparticles, the convergence criteria might be different. The conclusions are valid for other nanoparticles depending upon this criterion. The results of this work can be used in several industrial and engineering applications like in microelectronics, fuel cells, hybrid-powered engines nuclear reactors, transportations, biomedicine/ pharmaceutical processes and pasteurization of food etc.

Can you change your conclusions by adding this response?

Response: There is no need to change the conclusions, as we can use these conclusions for other nanoparticles.

Round 3

Reviewer 1 Report

ok